# A True-to-the-model Axiomatic Benchmark for Graph-based Explainers

**Corrado Monti**                                                                                       *corrado.monti@centai.eu*
*CENTAI Institute, Turin, Italy*

**Paolo Bajardi**                                                                                       *paolo.bajardi@centai.eu*
*CENTAI Institute, Turin, Italy*

**Francesco Bonchi**                                                                                  *francesco.bonchi@centai.eu*
*CENTAI Institute, Turin, Italy*

**André Panisson**                                                                                   *andre.panisson@centai.eu*
*CENTAI Institute, Turin, Italy*

**Alan Perotti**                                                                                       *alan.perotti@centai.eu*
*CENTAI Institute, Turin, Italy*

**Reviewed on OpenReview:** *https://openreview.net/forum?id=HSQTv3R8Iz*

## Abstract

Regulators, researchers, and practitioners recognize the urgency of explainability in artificial intelligence systems, including the ones based on machine learning for graph-structured data. Despite the large number of proposals, however, a common understanding of what constitutes a *good* explanation is still lacking: different explainers often arrive at different conclusions on the same problem instance, making it hard for practitioners to choose among them. Furthermore, explainers often produce explanations through opaque logic hard to understand and assess – ironically mirroring the black box nature they aim to elucidate.

Recent proposals in the literature for benchmarking graph-based explainers typically involve embedding specific logic into data, training a black-box model, and then empirically assessing how well the explanation matches the embedded logic, i.e., *they test truthfulness to the data*. In contrast, we propose a *true-to-the-model axiomatic framework* for auditing explainers in the task of node classification on graphs. Our proposal hinges on the fundamental idea that an explainer should discern if a model relies on a particular feature for classifying a node. Building on this concept, we develop three types of white-box classifiers, with clear internal logic, that are relevant in real-world applications. We then formally prove that the set of features that can induce a change in the classification correctly corresponds to a ground-truth set of predefined important features. This property allows us to use the white-box classifiers to build a testing framework.

We apply this framework to both synthetic and real data and evaluate various state-of-the-art explainers, thus characterizing their behavior. Our findings highlight how explainers often react in a rather counter-intuitive fashion to technical details that might be easily overlooked. Our approach offers valuable insights and recommended practices for selecting the right explainer given the task at hand, and for developing new methods for explaining graph-learning models.

# 1 Introduction

The success of machine learning methods in solving problems over graph-structured data has spurred a great deal of applications in a variety of domains, ranging from biology to finance, from social media to neuroscience. However, these models – and in particular graph neural networks (GNNs) – are often black boxes, as they have no intrinsic mechanism to provide human-understandable explanations of their inner decision logic (et al., 2020). Increasing concerns from regulators, practitioners, algorithmic-assisted decision-takers, and subjects of algorithmic decisions (e.g., patients, customers etc.) boosted the research effort devoted to the development of novel explainability techniques for graph classification (see Yuan et al. (2020) for a recent survey). Regardless of the great deal of attention, a common understanding of what makes a good explanation is still lacking. As a consequence, different explainers often arrive to completely different conclusions on the same problem instance. Moreover, those explainers are themselves, more often than not, black-boxes producing explanations through opaque logic hard to understand and assess. Because of these reasons, while the usage of explainers for tabular data is now common (e.g., SHAP, Lundberg & Lee (2017)), their adoption in practice for node classification in graphs is lagging behind: domain experts have no clear way to understand whether an explanation can be reliable, or to decide which explainer would be appropriate for a specific problem.

**Motivating example.** Let us consider the following scenario. Imagine we have a model running in production in a banking system, that employs the network of financial transactions, together with attributes of its users, to determine a risk score for each user. The role of an explainer in this setting might be to assert the user's *right to explanation* as required by EU regulations (Selbst & Powles, 2018; Goodman & Flaxman, 2017). In particular, a user might want to know if a particular protected feature (e.g. ethnicity or gender) of a node, or of its neighbors, is employed by the model in making an automated decision. A watchdog might analyze such a model using an explainer. Our goal is therefore to ensure that a given explainer can correctly diagnose whether a model is using a feature or not. While this task might be straightforward in tabular data, in node classification in graphs a specific feature of a node (e.g. ethnicity) might not be used directly to classify it, but it is still possible that the feature of its neighbors might be employed.

**Proposed framework.** In this work, we propose a rigorous framework aimed at assessing whether a node-classification explainer can identify when a given model uses a certain feature to produce the classification. Recent efforts in this direction (Rathee et al., 2022; Longa et al., 2022; Amara et al., 2022; Jaume et al., 2021) typically adopt the following approach: they (1) implant some target patterns in a synthetic training set, (2) train a black-box node classifier, (3) apply the explainer under scrutiny, (4) finally measure empirically the adherence of the produce explanation to the implanted logic, i.e., *they test truthfulness to the data.* Such an approach to auditing explainers in inherently complex, as it aims at assessing a software (the explainer), which in turn analyzes another software (the model), which in turn is the output of another software (the training algorithm). This process has many dependencies; for instance, it depends on the specific learning algorithm and its parameters. All these dependencies can make the overall auditing process unstable. Moreover, there is a degree of uncertainty about whether the trained black box has learned the intended logic (i.e., the planted patterns). Departing from this literature, we propose an axiomatic framework aimed at assessing whether a given explainer is *true to the model* (Chen et al., 2020): the explainer should capture the logic underlying the classification performed by the model, and not the patterns planted in the data. In fact, although present in the training data, those patterns might not be used by the classifier. Our goal is thus to evaluate the accordance of the explainer with the model itself, without making assumptions on what the model might have learned from the data. By measuring the concordance of explainers directly with the model logic, we can guarantee that our measure is not affected by any complex interaction between features in the data.

Our goal is formalized in a simple axiom: explainers should be able to differentiate whether a feature is used by a given model or not. Towards the operationalization of this intuition, we adopt a simple definition of what an *important* feature is within our framework. We define a feature $f$ *important* for the classification of a node $v$ by a model $\mathcal{M}$, if and only if changing it, while maintaining all the other features unchanged, changes the classification of $v$. By no means this can be considered a general definition of feature importance in machine learning, however, within the strict limits of our framework we consistently use "important" with

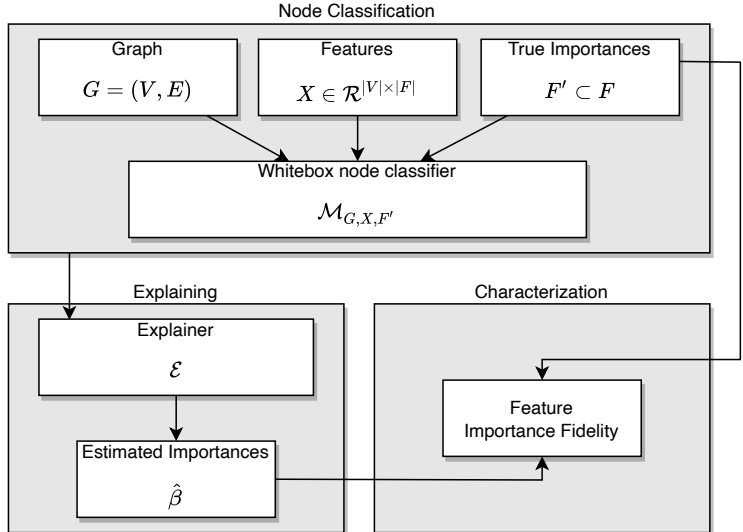

Figure 1: Our proposed framework. We start from a graph $G$ with node features. We randomly mark subsets of features ($F'$) as important: this is the ground truth for our evaluations. We also use a white-box node classifier $\mathcal{M}$ implementing a known logic, with ground truth importance for the features. We then feed these components to the explainer $\mathcal{E}$, thus obtaining an explanation as estimated importance for the features, which are then compared to the ground-truth.

this simple yet powerful semantic. We then axiomatize that an explainer should assign, to any important feature, a higher explanation score than to any unimportant feature.

We then devise three *"white-box models"*, i.e., classifiers whose internal logic is visible, in which we are able to hard-code which features are "important". Our white boxes have different logic and cover relevant scenarios in real-world applications of node classification: a *local model*, where the class of a node depends on a subset of its own features; a *neighborhood model*, where the class depends on the features of some of its neighbors; and a *two-hop model* where the class depends on the features of the nodes that are more likely to be reached in a two-hop random walk. All these models can also keep in consideration the degree of a node, which is often valuable for applications. For each white box we justify its practical relevance and we discuss its PyTorch implementation as message-passing GNN. The simplicity of our white boxes allows us to prove (Theorem 1) that the set of features that are able to induce a change in the classification, correctly corresponds to a predefined set of important features. Thanks to this theorem, we can develop our quantitative benchmark which, taking as input a given explainer, verifies whether it respects our axiom. A schematization of our framework is given in Figure 1.

**Our findings.** We apply our framework to several state-of-the-art explainers, carefully selected to cover all the main classes of approaches in the literature. Our analysis uncovers important phenomena emerging in different explainers and highlights key differences in semantics across explainers that are often overlooked.

1. Our findings show that explainers based on *surrogates* can deliver an incomplete explanation when the surrogate they use does not faithfully represent the given model to explain. For instance, our framework detects that GraphLIME (Huang et al., 2022) is not able to model correctly the neighborhood model, where the class of a node depends on a feature of its neighbors.

2. Our framework also shines a light on peculiar behaviors that could be easily overlooked. For instance, we discovered that some explainers interpret feature importance as a difference from zero-valued features. When this is the case, zero-valued features cannot be, by definition, important.

3. Furthermore, some explainers define important features only as those that *positively* contribute to the classification of a node to the output class, while others consider both positive and negative contributions as important.

These crucial differences in semantics among explainers highlighted by our auditing framework, typically lead to completely different behaviors: without a proper systematization, they can be an obstacle to clarity. The goal of our auditing framework is exactly to help assess and clarify the behavior of explainable graph learning techniques, aiding practitioners in choosing explainability methods that can correctly expose the criticality of graph learning applications.

**Summary of contributions and paper roadmap.** The technical contributions of this work can be summarized as follows:

- We present a quantitative benchmark based on three different white-box classifiers with known inner logic, to assess if an explainer can recognize whether a model is using some features (Section 3). For each white box we discuss its PyTorch implementation as message-passing GNN. We prove (Theorem 1) that if an explainer respects our axiom, then its output must necessarily be in accordance with the white-box ground truth.

- We use our framework to audit five explainers, chosen as representatives of the taxonomic classes of node-level explainers identified by Yuan et al. (2020) (Section 4).

- We show how the potential limits diagnosed by our framework are a product of the internal logic of each explainer (Section 5), we thus provide suggestions for their users and for designers of novel explainers. We are also the first to adopt *salient deconvolutional networks* (Mahendran & Vedaldi, 2016), an explainer that was originally introduced to explain image classification, to explain node classification on graphs. Such a method performs perfectly on our benchmark.

- We provide code to let other researchers use our framework to test or to develop new graph explainers. Our code is available at `https://github.com/corradomonti/axiomatic-g-xai`.

Finally, a key contribution of this work is to show that a radically different approach to assessing explainers, based on testing trustfulness-to-the-model through white boxes with hard-coded logic, is possible and can produce important insights. Next section discusses the most relevant related literature.

## 2 Related Work

Graphs serve as fundamental structures in a myriad of real-world domains, and as a consequence graph neural networks (GNNs) have evolved as a cornerstone in machine learning. Notable state-of-the art approaches include Graph Convolutional Networks (GCNs)(Welling & Kipf, 2016), Graph Attention Networks (GATs)(Veličković et al., 2017), and GraphSAGE (Hamilton et al., 2017). The effectiveness of these models comes at the cost of a remarkable opacity, which is especially problematic in domains where understanding and trust in model predictions are crucial.

In response, several explainers for GNNs have been developed, using techniques like *gradient-based methods* (Integrated Gradients, Sundararajan et al. (2017)), *perturbation* analysis (GNNExplainer, Ying et al. (2019)), *surrogate* models (GraphLIME, Huang et al. (2022)), *decomposition* approaches (Layer-Relevance Propagation, Bach et al. (2015)), and *substructure-based methods* (GraphShap, Perotti et al. (2023), SubgraphX, Yuan et al. (2021)). In turn, these alternative solutions produce explanations which are hard to evaluate and compare, making it hard to decide which explainer would be appropriate for a specific problem.

This gap is particularly evident in the context of graph-structured data due to its complexity and diversity. Several frameworks and desiderata have been suggested for evaluating explanations (Langer et al., 2021; van der Waa et al., 2021; Rosenfeld, 2021; Amparore et al., 2021; Agarwal et al., 2022a), but these often do not cater specifically to graph-based models, failing to capture their unique properties. The need for graph-specific evaluation frameworks is underscored in the taxonomic survey by (Yuan et al., 2020), which, while focusing on explainability methods for GNNs, does not address the auditing problem in depth.

Auditing efforts, such as those by Agarwal et al. (2022b), provide valuable insights but often rely on synthetic or empirical data. This reliance on approximations can lead to potential discrepancies between the actual and

perceived effectiveness of explainers. Other proposals are strongly domain-specific: for instance, Jaume et al. (2021) propose a set of quantitative metrics based on class separability statistics using pathologically relevant concepts, thus quantifying the alignment of algorithmically-generated explanations with known concepts in pathology. Besides the impossibility of generalising these metrics to other domains, these approaches highlight the lack of existing quantitative metrics for evaluating graph-based explanations. Other recent effort[1] is devoted to develop benchmarks for the evaluation of graph-based explanation. BAGEL (Rathee et al., 2022) focuses both on explanation *usability*, measuring explanation sparsity and plausibility, and on explanation *correctness*, through the metrics of faithfulness (ability to characterize model's inner logic) and correctness (ability to recognize the externally injected correlations). Longa et al. (2022) propose a similar comparative survey, where the implemented metrics are *plausibility* (consistency between the explainer mask and ground truth mask) and *fidelity* (consistency between the model prediction on the full graph and on the explanation subgraph). Remarkably, edge masks are converted to node masks. Finally, Amara et al. (2022) introduce the *characterization score*, combining necessary-based and sufficient-based fidelity metrics.

As already highlighted in the Introduction, all these approaches train black-box models, such as GCN and GAT, and fundamentally root their metrics on querying these models on perturbed and/or masked data, i.e., they test truthfulness to the data. Interestingly, in all the mentioned papers, the authors either explicitly acknowledge the strong limitation of assuming that the explanation ground truth is actually picked up by the model to make its decision or constrain their analysis to scenarios where the model almost memorized the injected patterns to achieve almost perfect classification. Conversely, we propose a true-to-the-model axiomatic benchmark by means of white-box models with known internal logic, which allows a direct comparison between models and explanations, by-passing the approximation and uncertainty introduced by the training process.

## 3 A True-to-the-model Benchmark

We consider binary node classification on a node-attributed directed graph, defined over a set of nodes $V$ and features $F$. In this setting a model $\mathcal{M}$ can be seen as a function that takes a graph $G = (V, E)$ with $E \subseteq 2^{V \times V}$, a node-feature matrix $X \in \mathbb{R}^{|V| \times |F|}$, and a node $v \in V$, it returns the probability that $v$ belongs to the positive class:

$$\mathcal{M} : 2^{V \times V} \times \mathbb{R}^{|V| \times |F|} \times V \to [0, 1].$$

Note that the graph $G$, being always defined over the same set of nodes $V$, is completely identified solely by the edge set $E \in 2^{V \times V}$. For simplicity, we omit $E$ and $X$ from $\mathcal{M}(E, X, v)$, when clear from the context, and just denote $\mathcal{M}(v)$ the outcome of model $\mathcal{M}$ on node $v$.

In this setting, an instance-level explainer is a function that takes the model $\mathcal{M}$, the edge set $E$, the node-feature matrix $X$, a node $v$ and returns an explanation for $\mathcal{M}(v)$, i.e., a score of importance for each feature $f \in F$. Again, we omit $E$ and $X$ when clear from the context and just denote the explainer as a function $\mathcal{E}_{\mathcal{M}} : V \to \mathbb{R}^{|F|}$. Here $\mathcal{E}_{\mathcal{M}}(v) = \beta$ denotes the explanation for $\mathcal{M}(v)$, where the vector $\beta$ determines the explanation scores: $\beta_f$ is the explanation score given to the feature $f \in F$.

### 3.1 The axiom: important features should receive higher importance score than non-important ones

We start a simple, yet fundamental, intuition that we want the explainers to respect: features truly employed by the model to perform a classification, should receive higher explanation scores than features that are not used. We thus adopt a simple definition of what an *important* feature is within our framework: a feature $f$ *important* for the classification of a node $v$ by a model $\mathcal{M}$, if and only if changing it, while maintaining all the other features unchanged, changes the classification of $v$.

**Definition 1.** *Given the graph $G = (V, E)$ with node-feature matrix $X \in \mathbb{R}^{|V| \times |F|}$, we say that a feature $f^* \in F$ is* important *for the prediction of a model $\mathcal{M}$ on node $v \in V$ iff it exists another matrix $Y \in \mathbb{R}^{|V| \times |F|}$, which is identical to $X$ except for the column $f^*$ (i.e., $y_{u,f} = x_{u,f}, \ \forall f \neq f^*, u \in V$) such that $\mathcal{M}(X, v) \neq \mathcal{M}(Y, v)$. We denote $F^*(v) \subseteq V$ the set of important features for the classification of $v$.*

---

[1]To the best of our knowledge, not yet peer-reviewed.

When clear from the context, we denote the set of important features for a node $v$, simply by $F^*$. We can now formalize our starting intuition in an axiom of the desired behavior of explainers w.r.t. important features. Recall that $\beta_f$ denotes the importance score given by the explainer to the feature $f \in F$.

**Axiom 1.** *Let $\mathcal{E}_\mathcal{M}$ be an explainer producing importance scores $\mathcal{E}_\mathcal{M}(v) = \beta$ when given model $\mathcal{M}$ and node $v$. Let $F^*$ denote the set of important features (Definition 1). It must hold that $\beta_f < \beta_{f^*} \ \forall f \notin F^*, f^* \in F^*$.*

We next introduce a benchmark of three simple models aimed at stressing a given explainer w.r.t. Axiom 1. Such models are *"white boxes"* (in contrast to black boxes) meaning that their internal logic is visible. In particular, they are geared on a predefined (ground truth) set $F' \subset F$ of important features, which is encoded inside each model as binary vector $\hat{\beta} \in \{0,1\}^{|F|}$ where $\hat{\beta}_f = \mathbf{1}_{F'}(f)$. After introducing such white boxes (Section 3.2), we prove that, for each of them, the ground truth set $F'$ and the set of important features coincide (Theorem 1). We finally (Section 3.3) measure how much the explanation $\beta$ produced by the explainer adheres to the ground truth $\hat{\beta}$, or in other terms, to which extent the explainer respects Axiom 1. It is worth stressing that our test is a necessary, but not sufficient, condition for an explainer to be considered a valid tool: passing our test, does not imply that an explainer might not have other issues, on the other hand, failing our simple test clearly highlight some important drawbacks of the given explainer.

## 3.2 The white-box models

We first provide a general schema common to all three white boxes, then we introduce them in details. For each of them, we discuss ($i$) their relevance to real-world applications of node classification, and ($ii$) their PyTorch implementation as message-passing graph neural networks.

As pointed out earlier, each white box receives, as part of the input, the set $F' \subset F$ of ground-truth features which is encoded as a binary vector $\hat{\beta}$. Inside each white-box model, $\hat{\beta}$ is used as a kernel: i.e., instead of considering the feature vector $x_v$ of a node $v$, the model considers $\hat{\beta}^\top \mathbf{x}_v$ (indicating with $\mathbf{x}_v$ the $v$-th row of $X$). The general schema for the white-box models $\mathcal{M}(E, X, v)$ is as follows:

$$\mathcal{M}(E, X, v) = \phi_{K,\vartheta}(\widehat{\mathcal{M}}(E, X, v)) \tag{1}$$

where $\widehat{\mathcal{M}}$ is a base model and $\phi_{K,\vartheta}$ is the sigmoid function

$$\phi_{K,\vartheta}(z) = \left(e^{K(\vartheta-z)} + 1\right)^{-1}. \tag{2}$$

The sigmoid function serves two purposes: first, it allows to obtain $y \in [0,1]$ even if the base model $\widehat{\mathcal{M}}$ has a range in $\mathbb{R}$; second, it normalizes such scores as explained next.

Consider $S$ as the sequence of outputs $\widehat{y} = \widehat{\mathcal{M}}(E, X, v)$ of the base model $\widehat{\mathcal{M}}$ for all the nodes $v \in V$. We use $S$ to adjust the midpoint $\vartheta$ and steepness $K$ of the sigmoid function $\phi_{K,\vartheta}(z)$. Specifically, we set $\vartheta$ as the mean of $S$, and its steepness $K$ as the inverse of the standard deviation of $S$. This adjustment serves as a normalization mechanism, ensuring that the rescaled output effectively has zero mean and unit variance. By doing so, we standardize the distribution of the base model's outputs across different datasets and models, ensuring that on average, nodes have an equal probability of being classified into one class or the other and that the variance in scores is consistent, facilitating comparability and interpretability across different modelling contexts.

As in real-world scenarios, node classifiers might employ a subset of the edges for their classification, according to some internal criteria, and this selection might have an impact on the performance of the explainers, our white boxes consider a randomly chosen subset $E' \subset E$, obtained by including each edge with a $\frac{1}{2}$ probability.

We next present in detail three different base models $\widehat{\mathcal{M}}$, which differ in their usage of the information coming from neighborhood nodes. In the following $N(v)$ indicates the set of predecessor of $v$ which are connected to $v$ through an edge in $E'$: i.e., $N(v) = \{u \in V | (u, v) \in E'\}$.

**Local model (LM).** In this model, the class of a node $v \in V$ depends solely on its features $x_v$. However, we also introduce a dependence on the in-degree of $v$ controlled by a parameter $\gamma \in \mathbb{R}$. The model assigns

the probability of $v$ belonging to the positive class as

$$\widehat{\mathcal{M}}(E, X, v) = \gamma|N(v)| + \hat{\beta}^\top \mathbf{x}_v \tag{3}$$

The PyTorch implementation is straightforward. First, the in-degree component is obtained via a one-layer message-passing network where each message is a 1 and the aggregation function is the sum. Then, the output of this network is given to the main model that integrates the local parts (i.e., adding $\hat{\beta}^\top \mathbf{x}_v$ and passing the output to the sigmoid).

This model is the simplest one we consider, but it represents a variety of real-world situations. Information from local attributes of a node and from its degree might often be sufficient to identify popular users in social media (Hansen et al., 2010) or high-quality web pages (Upstill et al., 2003).

**Neighborhood model (NM).** In this model, the class of a node $v \in V$ depends on the features $x_u$ of the neighboring nodes $u \in N(v)$. Moreover, as in the previous model, we introduce a dependence regulated by $\gamma$ on the in-degree. The model is thus defined by

$$\widehat{\mathcal{M}}(E, X, v) = \gamma|N(v)| + \sum_{u \in N(v)} \frac{\hat{\beta}^\top \mathbf{x}_u}{|N(v)|}. \tag{4}$$

Intuitively, in this model, the classification of a node in the positive class depends on the average of the important features of that node's neighbors.

To implement this model, the main PyTorch module uses two one-layer message-passing networks. The first is the degree one, as in the local model previously explained. The second layer's message through $(u, v) \in E'$ is the feature vector $x_u$, while the aggregation function is the average. Then, these two components are aggregated according to Eq. (4).

This model represents a typical situation in several application scenarios in which nodes are classified based on their neighbors. For instance, in opinion mining on social media, it is common to classify the opinion of a user using their neighbor's observable features (Barberá, 2015), leveraging the assumption of homophily. More in general, GNN models composed by a single GraphSAGE convolution (Hamilton et al., 2017) can be implemented as a combination of a local and a neighborhood models, and have been used in node classification tasks for citation networks, social media, and graph classification for protein-protein interactions.

**Two-hop model (2HM).** Our third white-box model represents a generalization of the former neighborhood model to random walks of length 2. While this behavior could easily be generalized to longer paths, we restrict our attention to this case since it is by far the most common in practice (Tang et al., 2015). Let $\mathbb{P}^2_{V,E'}(v, u)$ indicate the probability that a random walk of length 2 starting in $v \in V$ ends up in $u \in V$ on the graph $(V, E')$.

Then, we define our model as

$$\widehat{\mathcal{M}}(E, X, v) = \gamma|N^2_{E'}(v)| + \sum_u \mathbb{P}^2_{V,E'}(v, u) \cdot \hat{\beta}^\top \mathbf{x}_u \tag{5}$$

where

$$N^2_{E'}(v) = \{u \in V \mid \mathbb{P}^2_{V,E'}(v, u) > 0\}. \tag{6}$$

In other words, this model classifies nodes as positive according to the important features of the nodes that can be reached with two steps on the graph. The implementation of this model involves two layers of message-passing networks for the degree component, and two for the feature aggregation, which finally are aggregated according to Eq. (5). Each single layer shares the same structure as the one used by NM. Node classification using random-walk-based analysis of node attributes has been well explored in the literature (Huang et al., 2019). Moreover, the most scalable methods typically leverage information coming from the two-hop neighborhood (Tang et al., 2015).

### 3.3 The auditing framework

Now that we have presented our white-box models, we are ready to present our axiomatic benchmark. The first needed step is to prove that the above white boxes indeed have the desired behavior; that is, the features $F' \subset F$ that are encoded as important inside each model (through $\hat{\beta}$), coincide with the important features $F^*(v)$ of an explanation $\mathcal{M}(E, X, v)$, for the classification of a node $v \in V$, as in Definition 1. This is proved in Theorem 1, whose formal proof is deferred to Appendix A.

**Theorem 1.** *Let $\mathcal{M}$ be one of our three white-box node classifiers (LM, NM, or 2HM) defined over a graph $G = (V, E)$ with no zero-degree nodes and node-feature matrix $X \in \mathbb{R}^{|V| \times |F|}$. Let $\mathcal{X}_f$ be the set of matrices that might differ from $X$ only in the feature $f$: $\mathcal{X}_f = \{X' \in \mathbb{R}^{|V| \times |F|} \, | \, \forall v \in V, \forall i \in F \, . \, x'_{v,i} \neq x_{v,i} \Rightarrow i = f\}$. For the classification $\mathcal{M}(E, X, v)$ with $v \in V$, it holds that*

$$\hat{\beta}_f = 1 \Leftrightarrow \exists X' \in \mathcal{X}_f : \mathcal{M}(X', v) \neq \mathcal{M}(X, v),$$

*or in other terms, a feature $f$ is a ground-truth important feature $f \in F'$,* iff *changing it has the potential of changing the classification of $v$.*

Thanks to Theorem 1 we know that the three white boxes are such that the features $F^*$ (Definition 1) that should be recognized as important, correspond exactly to the ground-truth important features $F'$ that we can control. Thus, we can use them to measure how much the explanation $\beta$ produced by a given explainer adheres to the ground truth $\hat{\beta}$. We frame such a benchmark as a binary-classification task where the explainer, given a white box, needs to classify the features as important or not.

**Testing Explainers: The Feature Importance Fidelity.** Since Axiom 1 poses that $\beta_f < \beta_{f^*}$ for all $f \notin F^*, f^* \in F^*$, AUC ROC is the most natural measure, as it evaluates how well the scores $\beta$ can separate $F^*$ from $F \setminus F^*$. We take the mean value among all the test nodes. This measure that ranges from 0 to 1 is dubbed *feature importance fidelity*. A fidelity of 1 means that the explainer scores are perfectly able to separate the important features from the non-important ones. A fidelity of 0.5 corresponds to random performance: the importance scores are random with respect to the true importance of features. A fidelity of 0, instead, indicates anticorrelation: important features consistently receive lower importance scores than non-important ones.

To summarize, our framework, whose schematic representation is given in Figure 1, consists of the following steps.

1. Given a directed graph $G = (V, E)$ and a node-feature matrix $X \in \mathbb{R}^{|V| \times |F|}$, generate randomly a set of important features $F' \subset F$.

2. Using $F'$, define a white-box model $\mathcal{M}$, which treats as important features the set $F'$, encoded as $\hat{\beta}_f = \mathbf{1}_{F'}(f)$.

3. Run the explainer $\mathcal{E}$ on $\mathcal{M}$ for a set of test nodes $v \in V$, obtaining an importance score vector $\beta$, that should express (according to $\mathcal{E}$) the importance of each feature for the model $\mathcal{M}$ when classifying a node $v$.

4. For each test node, compute the ROC AUC between $\beta$ (the importance assigned by the explainer) and $\hat{\beta}$ (the ground truth importance of features for the white-box model $\mathcal{M}$). We call the mean of these measures the *feature importance fidelity*.

## 4 Experimental settings

In this section, we present the main external components we use in our experiments: explainers and graphs.

**Explainers.** In order to characterize a representative set of explainers, we choose one algorithm for each of the four classes identified in the survey by Yuan et al. (2020): gradient-based, perturbation, surrogate,

and decomposition. From the **Gradient-based** class, we select *Integrated Gradients* (Sundararajan et al., 2017), because of its implementation provided in Captum.[2] Albeit not included in the survey (Yuan et al., 2020) due to its recency, it belongs to this class as it uses the gradient values to approximate importance. Among the **Perturbation** explainers, we use *GNNExplainer* (Ying et al., 2019), the most widely-known representative of this class. Perturbation explainers, largely employed in image classification, analyze how an input perturbation impacts the model outputs in order to discover important features for the model. From the **Surrogate** class, we employ *GraphLIME* (Huang et al., 2022), the only method in this class which is able to assign importance to features, and for which an implementation exists.[3] As representative of **Decomposition** explainers, we select *Layer-wise Relevance Propagation* (Bach et al., 2015) (LRP). Its importance scores are computed by decomposing the final output of the model layer-by-layer according to layer-specific rules, until the input layer is reached.

Furthermore, we also consider a fifth method, **Deconvolution** (Mahendran & Vedaldi, 2016), an explainer that displays a hybrid decomposition - gradient-based approach. This method was originally introduced to explain image classification and, to the best of our knowledge, it has not been considered yet in literature as an explainer for node classification.

All of these methods share a common interface: they output an importance score for all features when given a model and a specific target node to classify.[4] As such, they fit in the framework summarized at the end of the last section.

**Datasets.** Although our white boxes are not trained, our framework still needs data. For our purposes, we use an array of Erdős–Rényi graph with 100 nodes and a set of random 50 binary features for each node. We generate an array of such data sets by varying the fraction of positive features. We opted for this kind of random graph in order to test the explainers on a networked system with topological features drastically different from the real data set. In our experiments, we randomly select a given fraction of features as important (varying among experiments) for the model $\mathcal{M}$, and we set the value of $\gamma$ to 1.

Finally, we also test a real-world dataset (He & McAuley, 2016) with 786 anonymized Facebook users (with 319 binary features) and 14024 edges between. Some features (e.g., gender) are considered protected for many tasks, and this data set has been widely used in studies of fairness in graphs (Dong et al., 2022). Our framework can therefore answer the question: *if a model $\mathcal{M}$ is using a protected feature $f$, for instance using the gender of a user to classify whether their ads should gain more visibility, is a given explainer $\mathcal{E}$ able to detect it?*

## 5 Results

We test the explainers presented above varying the values of important features and, for the synthetic data sets, the number of positive ones. We run each setting 4 times. From each experiment, we obtain a feature importance fidelity score that ranges from 0 (explainer scores are anticorrelated with important features) to 1 (all important features have a higher score than unused features, and thus Axiom 1 is respected). Values around 0.5 indicate random performance in capturing features' importance. We present our results on the array of synthetic data sets in Figure 2. In each plot, each dot represents the average fidelity obtained under each setting; whiskers extend to the maximum and minimum scores obtained. The different percentages of positive and important features are indicated on the x-axis of each plot. Figure 3 shows results for our framework on the Facebook data set; here, adherence to our axiom means that an explainer is able to detect whether the white-box models are employing a certain real, possibly protected, feature (such as the gender of users).

---

[2]https://captum.ai/

[3]`https://github.com/WilliamCCHuang/GraphLIME`: we double-checked that the implementation is correct with respect to the original paper (Huang et al., 2022).

[4]For some explainers where feature importance is reflected by score magnitude, we applied an absolute value function to their outputs, ensuring consistency with our framework's assumption of larger scores indicating more important features.

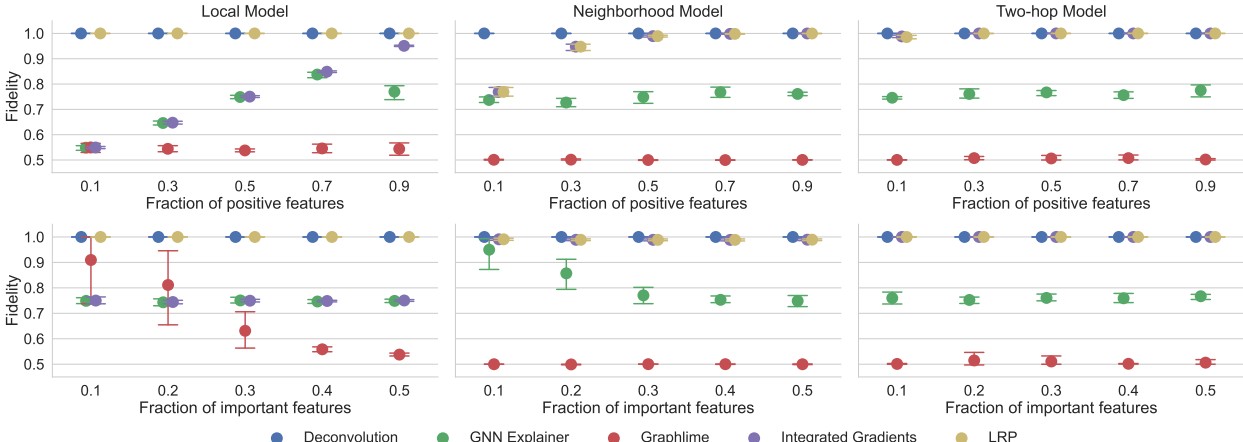

Figure 2: Feature importance fidelity measured by our framework on an array of synthetic data sets, with three white-box models to explain, one per column. Each explainer is represented by a different color. In each plot, the obtained fidelity is on the $y$-axis, and each dot represents the average of 4 experiments, whiskers extend to the minimum and maximum value obtained in each setting. In the top row, we vary the fraction of positive features (on the $x$-axis), and the fraction of important features is set to 50%. In the bottom row, we vary the fraction of important features, and the fraction of positive features is set to 50%.

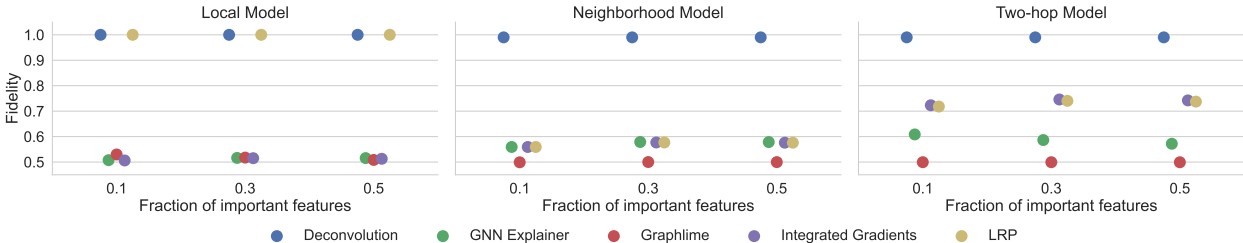

Figure 3: Feature importance fidelity measured by our framework on the Facebook users dataset with different fractions of randomly assigned important features, and three white-box models to explain, one per plot. In each plot, the fraction of important features is on the $x$-axis, and the obtained fidelity is on the $y$-axis. Each explainer is represented by a different color.

We next present these results explainer by explainer, highlighting under which conditions they respect Axiom 1, investigating the rationale beyond their performance, and discussing the consequences of their practical application.

## 5.1 GraphLIME

Among the explainers we assess, GraphLIME is the only one for which we are not able to find any setting where it fully respects Axiom 1. Its best results are obtained on the local model; in other contexts its performance often approaches a random baseline.

**Rationale.** These results show that GraphLIME suffers from the inability of its surrogate model to deal with non-local characteristics in node classification. To better understand the reasons, we have to analyze the way GraphLIME works. First, GraphLIME extracts a set of $n$ nodes in a $k$-hop subgraph around the explained node $i \in V$ ($k$ being a parameter, that we set to 2). Then, it considers the rows of the feature matrix $X$ corresponding to this set of nodes, and it computes a distance matrix $K \in \mathbb{R}^{|F| \times n \times n}$ between them: intuitively, the element $K_{i,j}$ represents a measure of similarity between $x_i$ and $x_j$ (we observe that this is computed for each pair of nodes $u$ and $v$ in the $k$-hop subgraph, regardless of whether $(u, v) \in E$ or

not). Following the same logic, GraphLIME computes a matrix $L \in \mathbb{R}^{n \times n}$ using the model's outcomes for each node: the element $L_{i,j}$ represents a measure of similarity between $y_i$ and $y_j$. Finally, GraphLIME uses these two matrices to explain predictions by using a regression method (i.e., its surrogate model): it assumes that if two nodes obtain a similar classification score when their feature $f$ is similar, then $f$ is important; otherwise, it is not.

We can see how this approach does not account for non-local correlations. Two nodes might have different features, but if *the features of their respective neighbors* are very similar, a message-passing model might assign them similar labels. In this case, no correlation would be detected by GraphLIME between the similarity of a specific feature and the model's outcome; even in the case that the outcome depends only on that feature. Moreover, we observe that even in the local model, GraphLIME obtains worse performances when there are many important features. This phenomenon is due to the regularization term in GraphLIME regression model, which favors sparse solutions.

**Fixes and warnings.** In the general case, GraphLIME might not be able to explain models that consider non-local features. Therefore, users should carefully evaluate if the regression model proposed by GraphLIME can suit their use case.

The performance of GraphLIME on our axiomatic benchmark highlights the defining bias of surrogate explainers: their simplified model can be incapable of explaining a model that follows a logic that cannot be modeled by the surrogate one. This hints at a promising direction for future research.

## 5.2 LRP

LRP respects Axiom 1 in the local model and on the two-hop model; on the neighborhood model, it does so only when the data set presents few positive features. As we will see next, this crucially depends on how we choose to represent negative features.

**Rationale.** The fundamental reason behind the scores obtained by LRP in our axiomatic benchmark, is that this method is not always able to correctly identify the zero-valued features as important. In the original paper Bach et al. (2015), in fact, authors normalize their image data set so that relevant features are not encoded as zeros. The importance of this step is nowhere acknowledged, for instance, it does not seem to be mentioned in the documentation of the implementation we use (https://captum.ai/api/lrp.html). If this fact is not clear to users, it might lead to an incorrect interpretation of LRP explanations.

Let us consider a practical example: opinion mining of social media users. Suppose to have a feature $f$ representing whether they have blocked a conservative politician account: 1 encodes them having blocked them and 0 they have not. If a model uses this feature to classify an individual $i$, when the feature $X_{i,f}$ is 0, the explanation offered by LRP will not assign high importance to that feature.

This issue is more present when there are more zero-valued features, and less impactful when there are more positive features, e.g., on the two-hop model than on the neighborhood one. Finally, since the possibility of a component being exactly zero greatly decreases when all the neighborhood features are averaged, this phenomenon is less severe when explaining non-local models. We observe that with LRP (contrarily to Integrated Gradient, as we will show below) this issue does not affect the local model but only the neighborhood one. We conjecture that this issue affects the propagation rule of the importance decomposition inside LRP: when the model is composed of only one layer–as in the local one–such an issue does not appear.

**Fixes and warnings.** Users should be warned that this explainer will not work as expected with zero-valued features. A simple solution is to choose a different encoding. For example, if we encode binary features with $\{-1, 1\}$ instead of $\{0, 1\}$, the feature importance fidelity of this explainer becomes almost perfect. We test this claim with an experiment, with the same exact setting as the one previously used, but encoding binary features with $\{-1, 1\}$ instead of $\{0, 1\}$. Figure 4 shows our results: as expected, LRP fidelity shifts from around 0.5 to close to 1 against all models, thus confirming our hypothesis.

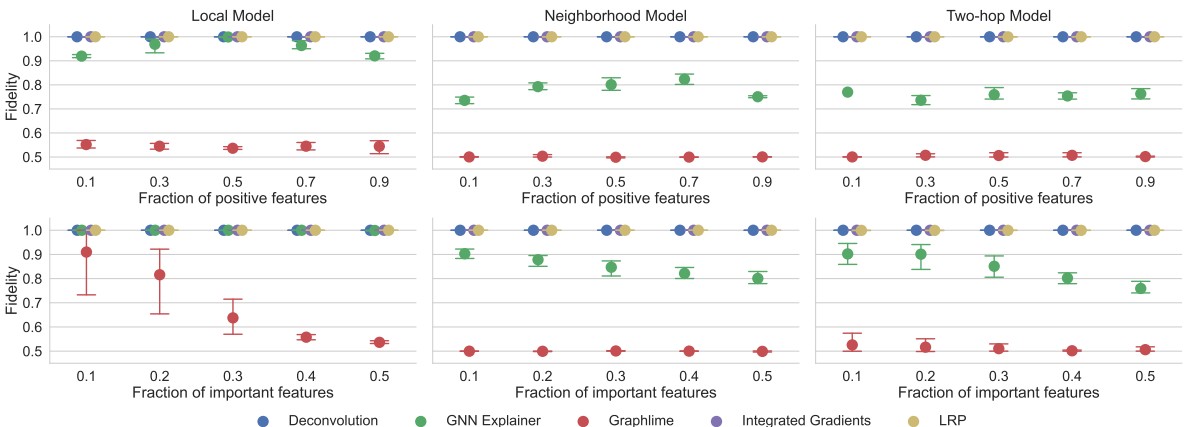

Figure 4: Feature importance fidelity when binary features are encoded with $\{-1, 1\}$ instead of $\{0, 1\}$, for each of the three white-box models (one per column). In each plot, the fidelity is on the $y$-axis. In the top row, the $x$ axis reports different fractions of positive features; in the bottom row, it reports different fractions of important features.

## 5.3 Integrated Gradients

Integrated Gradients shows in general results similar to LRP. The only difference lies in the local model in the case features are encoded as zero, where Integrated Gradients obtains worse results than LRP.

**Rationale.** Integrated Gradients suffers from the same issue of LRP when important features are encoded as zeroes: this phenomenon for Integrated Gradients is very clear even in the local model, where fidelity scales proportionally with the fraction of positive features. This behavior confirms that zero-valued features do not behave as expected with this explainer. We dub explainers with this issue as *zero-baseline explainers*: in fact, Integrated Gradients' authors Sundararajan et al. (2017) explicitly state this requirement, observing that the explanation depends on a *baseline input* that *"for image networks, [..] could be the black image, while for text models it could be the zero embedding vector. [..] A key step in applying integrated gradients is to select a good baseline. We recommend that developers check that the baseline has a near-zero score."*

**Fixes and warnings.** As for LRP, Figure 4 shows that by encoding binary features with $\{-1, 1\}$ instead of $\{0, 1\}$, Integrated Gradients' performance shifts to a perfect one.

## 5.4 GNNExplainer

We observe that GNN Explainer obtains a fidelity score proportional to the number of positive features when they are encoded as $\{0, 1\}$ (similarly to Integrated Gradients). With non-local models, instead, it obtains fair yet not-perfect results (fidelity $\sim 0.75$), regardless of the type of encoding. This problem is alleviated (fidelity $\sim 0.8 - 0.9$) when the important features are fewer.

**Rationale.** First of all, we recognize that also GNN Explainer is unable to recognize important features when their value is zero; as such, we focus on the experiments where features are encoded as $\{-1, 1\}$. We show such results in Figure 4. Here, the imperfect results obtained by this explainer are due to a combination of different factors. First, we find that GNNExplainer has a different semantic than other explainers. To better explain this semantic difference we use the notation introduced in Section 3. We consider a model $\mathcal{M}$ that takes as input the features $X$ and classifies a node $v$ as class $c \in \{0, 1\}$, in particular $c = 1$ when $\mathcal{M}(X, v) > \vartheta$ for some threshold $\vartheta \in [0, 1]$. An explainer $\mathcal{E}$ analyzing this classification instance will output $\mathcal{E}_{\mathcal{M}}(X, v) = \beta$, assigning an importance score $\beta_f$ to each feature $f$. Now, such scores can obey one of the two distinct semantics that we define next.

(1) *Signed attribution.* Importance scores $\beta$ should identify all features that have any impact, either positive or negative, in the classification of the node $v$ as class $c$. This is equivalent to what is required by Axiom 1.

(2) *Decision-aligned attribution.* Importance scores $\beta$ should identify only features that move the classification of the node $v$ towards the actual class $c$. In other words, $\beta_f$ should be high when changing $x_{v,f}$ increases $\mathcal{M}(X, v)$ in the case $\mathcal{M}(X, v) > \vartheta$, or decreases $\mathcal{M}(X, v)$ in the case $\mathcal{M}(X, v) < \vartheta$.

While other explainers implement the former semantic, GNNExplainer follows the latter: only features that reinforce the given class assignment are deemed as important.

Even besides this semantic difference, the results obtained by GNNExplainer are not perfect. Upon additional experiments, we observe that the presence of a mask on the edges significantly impacts the performance of GNNExplainer. In other words, the presence of edges where varying a feature does not affect the output (even when there are edges where it does) makes it harder for this explainer to recognize a feature as important. Finally, the constraint imposed by default by GNNExplainer to select a low number of important features often leads to a lower number of features recognized as important. For this reason, its performance degrades when there is a high number of important features.

**Fixes and warnings.** Regarding semantics, while both semantics have their applications, we emphasize the importance of distinguishing the two, so that practitioners can understand correctly the obtained explanations.

Regarding sparsity, we find out that GNNExplainer is very sensitive to some hyperparameters, such as $K_M$ and $K_F$ that control the size of subgraph and feature explanations respectively. In this work the default values of hyperparameters are used across all experiments, i.e., the regularization hyperparameters for subgraph size is 0.005 and for feature explanation is 1. Optimal values should be informed by prior knowledge about the dataset, and as can be seen from experiments, fixed values of hyperparameters might result in degraded performance if they are not optimal for the number of features in the ground truth.

### 5.5 Deconvolution

In every configuration, Deconvolution obtains perfect results with a fidelity of 1.0, showing that it is the only explainer to fully adhere to Axiom 1 in all tested settings. Its perfect results, however, do not come for free. Since it is not model-agnostic, its application is fundamentally limited to graph neural networks. Also, it might require modifications at the implementation level or in the logic of the classifier in order to be applicable. However, the reliability of the explanations offered by Deconvolution for graph machine learning models constitutes a promising and novel result.

## 6  Discussion

This work proposes an axiomatic approach to audit node classification explainers through a model-driven framework. In particular, our proposal is to assess whether an explainer is able to detect as important, features that are hard-coded as such, in three white-box models with known internal logic. The simplicity of the three white boxes that we define allows us to prove that the features that exhibit the capability of changing the classification, correctly correspond to the hard-coded set of ground-truth important features. Therefore, we can directly assess the concordance of explainers with the model logic, avoiding the training of a black-box on some syntectic data as done by the other proposals in the literature. The main advantage of our approach over this literature lies exactly in bypassing the training phase with its inherent uncertainty about whether the intended logic implanted in the training data has been indeed maintained in the learned model. Moreover, one main contribution of our paper is showing that this different approach, assessing trustfulness-to-the-model, is indeed possible and can produce interesting insights.

We applied our framework to five explainers, covering the main classes of approaches in the literature. Our analysis brought to light the existence of some subtle differences among explainers that have an important impact on the quality of the explanations and that are easily overlooked.

| Model Name | GNNExplainer |
|---|---|
| **Intended Use Case** | The explainer is intended to be used to gain insights into how GNN models make predictions, and to increase transparency and trust in the predictions made by these models. |
| **Model Aware/Agnostic** | The algorithm is defined as model agnostic, the available implementation is model (and library) specific. |
| **Zero Awareness** | GNNExplainer is a zero-baseline explainer, as it treats zero-valued features as a baseline: their importance might go undetected. Feature encoding is critical (e.g. use $\{-1, 1\}$ instead of $\{0, 1\}$ for binary features). |
| **Attribution Semantics** | A feature is considered important only if it contributes in the direction of the current classification. |
| **Employed Feature Detection** | Fully respected only when node classification is driven by local features. |

Table 1: Example of the contribution of the axiomatic framework to an explainer model card for GNNExplainer.

**Limitations.** Our contributions should be viewed within the limits of our work. Firstly, our framework follows a binary logic: it does not determine which explainers provide a better approximation of the different degrees of importance of each feature used by the model, but only which explainer is able to correctly identify important features, i.e., features that are used to make a prediction. It is worth noting that our simple definition of important features does not capture complex scenarios, e.g., a prediction that depends on a disjunction of important features. Yet, we show that this elementary criterion is nevertheless sufficient to develop a significant complexity in our analysis, bringing to light some important differences and pitfalls of different explainers. We recognize that our work only operationalizes one notion of truthfulness to the model, from which one metric is derived. However, our metric should not be the *sole* criterion for selecting among different explanation algorithms. Importantly, human aspects such as usability and adherence to the desired semantics must be considered in making such choices. In the light of these limitations, we consider our proposal as a first, yet important, step towards developing more nuanced axiomatic frameworks.

**Future work.** In our future work, we aim to tackle such limitations by extending our framework in several directions. First, as discussed above, we would like to generalize the framework to handle the notion of non-binary importance of features. Second, we aim to devise more complex white boxes, for instance where important features could be different on different nodes, or including non-linear transformations of the features: towards this goal, it is needed to strike a balance between the complexity of GNNs and the simplicity required to achieve formal guarantees as the ones provided by Theorem 1, whose extension to general GNNs is not easy. Third, we aim to generalize our framework to different tasks: e.g., graph classification or link prediction.

**Broader impact.** Besides these technical developments, we believe that a fundamental aim for the research community's future goals should be securing the reliability of *analyses* on machine learning systems. We hope that our framework represents a step in this direction, by allowing any researcher to use the proper tools to verify if a machine learning model is using or not a feature — possibly protected, such as their gender, religion, or their political ideas — to guide choices that could impact individuals. In this regard, our framework can also be an ingredient towards building a *model card for explainers*, similarly to what has

been done by Mitchell et al. (2019) for trained machine learning models. A model card provides information about a model for the sake of transparency and accountability for its creators and users, mitigating bias, fairness issues, and other ethical and power-balance concerns in their deployment. Similarly, an explainer model card should warn the user of an explainer on the limitations, the semantics, and the intended use of a given explainer—allowing practitioners to assess whether that explainer is fit for their particular purpose. In this sense, as exemplified in Table 1, our framework allows to check whether an explainer is fit to detect the usage of a given, possibly protected, feature in the context of graph-based machine learning. Building and analyzing explainers, and offering formal guarantees to their behavior, is a step towards making anyone able, in principle, to open the black boxes of key decision-making systems.

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

## A  Proof of Theorem 1

*Proof.* We proceed by proving each direction of this equivalence ($\Rightarrow$, $\Leftarrow$) for each of our three white boxes (LM, NM, 2HM).

**LM ($\Rightarrow$):** Assume $\hat{\beta}_f = 1$. We construct $X' \in \mathcal{X}_f$ as $x'_{v,f} = x_{v,f} + k$, $\forall v \in V$, for a constant $k \neq 0$. By definition of LM we have that $\mathcal{M}(X', v) = \phi(|N_{E'}(v)| + \sum_{i \neq f} \hat{\beta}_i x'_{v,i} + \hat{\beta}_f x'_{v,f})$ and by construction of $X'$, it follows $\mathcal{M}(X', v) = \phi(|N_{E'}(v)| + \sum_{i \neq f} \hat{\beta}_i x_{v,i} + \hat{\beta}_f x_{v,f} + \hat{\beta}_f k)$, which can be rewritten grouping the two terms $\phi(|N_{E'}(v)| + \hat{\beta}^\top \mathbf{x}_v + \hat{\beta}_f k)$. Since $\hat{\beta}_f = 1$, then $\mathcal{M}(X', v) = \phi(|N_{E'}(v)| + \hat{\beta}^\top \mathbf{x}_v + k)$. Now, since $k \neq 0$ and $\phi$ (as defined in Equation 2) is injective, we have

$$\mathcal{M}(X', v) = \phi(|N_{E'}(v)| + \hat{\beta}^\top \mathbf{x}_v + k) \neq \phi(|N_{E'}(v)| + \hat{\beta}^\top \mathbf{x}_v) = \mathcal{M}(X, v).$$

**LM ($\Leftarrow$):** Assume $\exists X' \in \mathcal{X}_f : \mathcal{M}(X', v) \neq \mathcal{M}(X, v)$. Now assume by absurd that $\hat{\beta}_f = 0$. By definition of LM we have that: $\mathcal{M}(X, v) = \phi(|N_{E'}(v)| + \hat{\beta}^\top \mathbf{x}_v) = \phi(|N_{E'}(v)| + \sum_{i \neq f} \hat{\beta}_i x_{v,i} + \hat{\beta}_f x_{v,f})$. As $\hat{\beta}_f = 0$, then $\mathcal{M}(X, v) = \phi(|N_{E'}(v)| + \sum_{i \neq f} \hat{\beta}_i x_{v,i})$. Now consider $X'$. It holds that

$$\mathcal{M}(X', v) = \phi(|N_{E'}(v)| + \sum_{i \neq f} \hat{\beta}_i x'_{v,i} + \hat{\beta}_f x'_{v,f}).$$

By definition of $\mathcal{X}_f$, we have that $\sum_{i \neq f} \hat{\beta}_i x'_{v,i} = \sum_{i \neq f} \hat{\beta}_i x_{v,i}$ and therefore

$$\mathcal{M}(X', v) = \phi(|N_{E'}(v)| + \sum_{i \neq f} \hat{\beta}_i x_{v,i} + \hat{\beta}_f x'_{v,f}).$$

As $\hat{\beta}_f = 0$, then $\mathcal{M}(X', v) = \phi(|N_{E'}(v)| + \sum_{i \neq f} \hat{\beta}_i x_{v,i}) = \mathcal{M}(X, v)$, $\forall X' \in \mathcal{X}_f$ which contradicts the initial assumption.

**NM ($\Rightarrow$):** Assume $\hat{\beta}_f = 1$. We construct $X' \in \mathcal{X}_f$ as $x'_{v,f} = x_{v,f} + k$, $\forall v \in V$, for a constant $k \neq 0$. By definition of NM we have that $\mathcal{M}(X, v) = \phi\left(\gamma|N(v)| + \frac{1}{|N(v)|} \sum_{u \in N(v)} \sum_{i \in F} \hat{\beta} x_{u,i}\right)$. For simplicity we rewrite it as $\mathcal{M}(X, v) = \phi(Q_1 + Q_2 + Q_3)$ where:

$$\begin{aligned} Q_1 &= |N(v)|; \\ Q_2 &= \frac{1}{|N(v)|} \sum_{u \in N(v)} \sum_{i \neq f} \hat{\beta} x_{u,i}; \\ Q_3 &= \frac{1}{|N(v)|} \sum_{u \in N(v)} \hat{\beta}_f x_{u,f}. \end{aligned}$$

Similarly we denote $\mathcal{M}(X',v) = \phi(Q'_1 + Q'_2 + Q'_3)$. We have that $Q_1 = Q'_1$ and $Q_2 = Q'_2$ due to our definition of $X' \in \mathcal{X}_f$. For what concerns $Q'_3$ instead we have that

$$
\begin{aligned}
Q'_3 &= \frac{1}{|N(v)|} \sum_{u \in N(v)} \hat{\beta}_f x'_{u,f} \\
&= \frac{1}{|N(v)|} \sum_{u \in N(v)} \hat{\beta}_f x_{u,f} + \frac{1}{|N(v)|} \sum_{u \in N(v)} \hat{\beta}_f k \\
&= Q_3 + \frac{1}{|N(v)|} \sum_{u \in N(v)} \hat{\beta}_f k.
\end{aligned}
$$

Since $\hat{\beta}_f = 1$, $k \neq 0$ and $N(v)$ is not empty (no singleton nodes), it follows that $\sum_{u \in N(v)} \hat{\beta}_f k \neq 0$. Therefore, since $\phi$ is injective, $\phi(Q_1 + Q_2 + Q_3) \neq \phi(Q_1 + Q_2 + Q_3 + \sum_{u \in N(v)} \frac{\hat{\beta}_f k}{|N(v)|})$ and $\mathcal{M}(X,v) \neq \mathcal{M}(X',v)$.

**NM ($\Leftarrow$):** Assume $\exists X' \in \mathcal{X}_f : \mathcal{M}(X',v) \neq \mathcal{M}(X,v)$. Now assume by absurd that $\hat{\beta}_f = 0$. Using the same rewriting as above we denote $\mathcal{M}(X,v) = \phi(Q_1 + Q_2 + Q_3)$ and $\mathcal{M}(X',v) = \phi(Q'_1 + Q'_2 + Q'_3)$, with $Q_1 = Q'_1$ and $Q_2 = Q'_2$ as the case above. Instead, contrarily to the case above, since we are assuming that $\hat{\beta}_f = 0$, we also have that $\forall X' \in \mathcal{X}_f$, $Q_3 = Q'_3$ and thus $\mathcal{M}(X,v) = \mathcal{M}(X',v)$, $\forall X' \in \mathcal{X}_f$, which is a contradiction with the assumption.

**2HM ($\Rightarrow$):** Assume $\hat{\beta}_f = 1$. We construct $X' \in \mathcal{X}_f$ as $x'_{v,f} = x_{v,f} + k$, $\forall v \in V$, for a constant $k \neq 0$. By definition of 2HM we have that $\mathcal{M}(X,v) = \phi(|N^2_{E'}(v)| + \sum_u \mathbb{P}^2_{V,E'}(v,u) \cdot \hat{\beta}^\top \mathbf{x}_u)$. For simplicity we rewrite it as $\mathcal{M}(X,v) = \phi(Q_1 + Q_2 + Q_3)$ where:

$$
\begin{aligned}
Q_1 &= |N^2_{E'}(v)|; \\
Q_2 &= \sum_u \sum_{i \neq f} \mathbb{P}^2_{V,E'}(v,u) \cdot \hat{\beta}_i x_{u,i}; \\
Q_3 &= \sum_u \mathbb{P}^2_{V,E'}(v,u) \cdot \hat{\beta}_f x_{u,f}.
\end{aligned}
$$

Similarly we denote $\mathcal{M}(X',v) = \phi(Q'_1 + Q'_2 + Q'_3)$. For all $X' \in \mathcal{X}_f$, we have that $Q_1 = Q'_1 = |N^2_{E'}(v)|$ and $Q_2 = Q'_2$. Instead, $Q'_3$ is

$$
\begin{aligned}
Q'_3 &= \sum_u \mathbb{P}^2_{V,E'}(v,u) \cdot \hat{\beta}_f x'_{u,f} \\
&= \sum_u \mathbb{P}^2_{V,E'}(v,u) \cdot \hat{\beta}_f x_{u,f} + \sum_u \mathbb{P}^2_{V,E'}(v,u) \cdot \hat{\beta}_{f,k} \\
&= Q_3 + \sum_u \mathbb{P}^2_{V,E'}(v,u) \cdot \hat{\beta}_{f,k}.
\end{aligned}
$$

As the graph has no zero-degree nodes the random walk is well-defined and $\exists u \, \mathbb{P}^2_{V,E'}(v,u) > 0$. Moreover, we assumed $\hat{\beta}_f = 1$ and $k \neq 0$, and thus $\sum_u \mathbb{P}^2_{V,E'}(v,u) \cdot \hat{\beta}_f k \neq 0$. Therefore, since $\phi$ is injective, $\phi(Q_1 + Q_2 + Q_3) \neq \phi(Q_1 + Q_2 + Q_3 + \sum_u \mathbb{P}^2_{V,E'}(v,u) \cdot \hat{\beta}_f k)$ and $\mathcal{M}(E,X,v) \neq \mathcal{M}(E,X',v)$.

**2HM ($\Leftarrow$):** Follows exactly the proof of the case NM ($\Leftarrow$).

This concludes the proof. $\qquad\qquad\qquad\qquad\qquad\qquad\qquad\qquad\qquad\qquad\qquad\qquad\qquad\qquad\quad$ $\square$

