# OpenReview forum: "A True-to-the-model Axiomatic Benchmark for Graph-based Explainers"
_TMLR — Accepted by TMLR_

### Review · Reviewer_MeAS · 2024-02-16

**Summary Of Contributions:**

In this research, the authors proposed a method to evaluate the faithfulness of feature-based explanations provided for Graph Neural Networks (GNNs). This method aims to assess whether these explanations effectively capture the truly important features in the model.
Specifically, they constructed simple GNNs that encapsulate the truly important features. Then, they applied various explanation methods to these simple GNNs to determine how accurately they could identify the truly important features.
The authors conducted evaluations on representative explanation methods such as Integrated Gradient, GNNExplainer, GraphLIME, LRP, and Deconvolution, demonstrating experimentally that many of these methods could not accurately identify the true important features with sufficient accuracy.
Furthermore, they provided qualitative and quantitative insights into the reasons behind the low accuracy of these methods.
The authors also reported that Deconvolution exhibited the most promising results in this evaluation.

**Audience:**

Yes

**Broader Impact Concerns:**

There is no ethical concern.

**Claims And Evidence:**

Yes

**Requested Changes:**

Most of the followings are the questions to the authors rather than changes.

#### Request 1:
The proposed GNNs all operate under the assumption of globally important features. Does this assumption pose any risk of bias in evaluating explanation methods? Would it be more appropriate to design GNNs based on the assumption of locally important features and evaluate on them?

#### Request 2:
In the proposed evaluation method, simple classical techniques like Gradient or Permutation Importance are trivially effective. Does this fact indicate the effectiveness of these classical methods, or does it suggest a missing crucial aspect in this evaluation method?

#### Request 3:
It might be more beneficial to remove Theorem 1 and allocate pages to expanding experiments, analyzing their results, and providing insights rather than proving obvious theorems.

**Strengths And Weaknesses:**

### Strong aspects

#### Strength: Impact of Results
The authors have highlighted that typical explanation methods such as Integrated Gradient, GNNExplainer, GraphLIME, and LRP tend to fail to identify important features in simple GNNSs. This finding suggests the need for careful consideration of the reliability of these methods when applied in practical scenarios.
Furthermore, the authors revealed that in methods based on zero baselines like Integrated Gradient and LRP, accuracy improvement is possible by transforming features to become non-zero.
Insights gained from these discoveries and improvements can be considered an important step forward in Explainable Artificial Intelligence (XAI) research.

### Weak aspects

#### Weakness 1: Global vs. Local Feature Importance
The proposed simple GNNs operate under the assumption that important features are consistent across all nodes. This implies that the evaluation method assumes GNNs make decisions based on globally important features consistently across all decisions. However, a key advantage of Neural Network-based models is their ability to adaptively change decisions based on different features depending on the data (nodes). Therefore, evaluating GNNs that make decisions based on locally important features, where important features vary for each data (node), would better suit real-world problem settings. Thus, the proposed evaluation model oversimplifies the situation and seems to overlook the scenario where the important features assumed by many explanation methods vary locally.

#### Weakness 2: Existence of Trivial Solutions
The proposed GNNs (3), (4), (5) are sparse linear models. Consequently, it's easy to identify important and non-important features by perturbing the input vector x. For instance, a simple gradient, i.e., differentiating the model's output with respect to the input vector x, makes non-important features to have zero derivatives, easily achieving Axiom 1. Similarly, with popular explanation methods like Permutation Importance, permutations for non-important features don't alter the model's output, hence their contribution can be deemed zero. In the proposed evaluation method, these classical and extremely simple explanation methods can trivially achieve the highest accuracy. Does this fact suggest the effectiveness of these classical methods, or does it imply a missing crucial aspect in this evaluation method?

#### Weakness 3: Theorem 1 is trivial.
Theorem 1 seems trivial based on the definition of the models (3), (4), (5). Devoting many pages to demonstrating this trivial result doesn't seem meaningful. It would be more beneficial to allocate pages to expanding experiments, analyzing their results, and providing insights rather than proving obvious theorems.

---

> ### Author Response · Authors · 2024-02-28
> **Response from Authors**
>
> We would like to thank the Reviewer for their helpful comments, which have allowed us to improve our work.
> We also appreciate that the Reviewer recognizes the impact of our work.
> We next discuss the 3 weaknesses raised by the Reviewer (and the 3 corresponding requests).
>
> ###  **[W1 and R1] (Global Vs. local feature importance)**
>
> We wholeheartedly agree with the reviewer that we are indeed considering a very simple setting. However, it is exactly the simplicity of the setting that allows us to develop a framework that is formally exact (that is, where our proof holds). This is a key distinction with the more heuristic approaches in the literature (see also response to [W2] of Reviewer GwDM). We show that our elementary criterion is nevertheless sufficient to develop a framework able to uncover some important differences and pitfalls of different explainers. In particular, in the case analyzed by the Reviewer, for the purposes of our framework the union of the sets of node-specific important features would be considered important. Anyway, we agree with the reviewer that it is important the reader is aware of this limitation: as such, we added it to our Future Works in Section 6.
>
> ###  **[W2 and R2] (Existence of trivial solutions)**
>
> Our goal is to equip real-world explainers with a sanity check able to verify if they respect a basic property in the explanations they produce. As such, we consider our criterion as a necessary, not sufficient, condition for explainers to be considered valid. Still, we show that some explainers do not pass this check. At the same time, it is expected that trivial simplistic explainers can satisfy our condition. Please refer also to  [W3] of Reviewer GwDM.
>
> ###  **[W3 and R3] (Theorem 1)**
>
> Thank you for this comment. We agree that the proof of Theorem 1 is rather straightforward. However, this does not reduce the importance of the Theorem which is a cornerstone of our framework. For this reason, in the first submission, we decided to keep its (formal and detailed) proof in the main body of the paper. Now we realize it wasn’t a great presentation choice. We have thus moved its proof to the Appendix.

---

> > ### Comment · Reviewer_MeAS · 2024-03-15
> > **Re: Response from Authors**
> >
> > I would like to thank the authors for the detailed reply.
> > I confirm that [W1] and [W3] are properly addressed in the revised version.
> > For [W2], I would like to suggest the authors for adding simple explainers that can trivially fulfill the axiom.
> > We can then find that these simple explainers are preferrable ones (in terms of the axiom), which also highlights the limiation of the propsoed evaluation metric.

---

> > > ### Author Response · Authors · 2024-03-15
> > > **Re: Re: Response from Authors**
> > >
> > > Thanks for your comment. In the revised version we will add the discussion about [W2] at the end of Section 3, after introducing the framework. We will highlight how trivial simplistic explainers can satisfy our condition. This in turn will help better stress the fact that our criterion is a necessary, not sufficient, condition for explainers to be considered valid.

---

### Review · Reviewer_STRH · 2024-02-17

**Summary Of Contributions:**

The paper introduces a framework for evaluating and auditing explainers used in graph-based models. They establish a benchmark using three white-box classifiers implemented as message-passing GNNs and prove a theorem ensuring explainer accuracy. The authors audit five explainers, identifying limitations and providing recommendations. They also adapt salient deconvolutional networks for graph node classification, performing perfectly on the proposed benchmark.

**Audience:**

Yes

**Broader Impact Concerns:**

N/A.

**Claims And Evidence:**

Yes

**Requested Changes:**

See weakness

**Strengths And Weaknesses:**

Strengths:

-	This paper introduces a framework for evaluating and auditing explainers used in graph-based models.

-	The framework is based on three white-box classifiers, implemented based on message-passing GNN and covering models relying on local features, one-hop and two-hop neighbors for node classification.

Weakness:

-	The framework is focused on auditing in terms of the feature importance. However, graph structure is also important for graph learning tasks, and it seems that the framework proposed in this paper cannot analyze the importance of graph structure.

-	Although three white-box classifiers can be implemented with message passing GNN, these white-box classifiers may differ from the classifiers learned from data. Therefore, it is necessary to discuss the differences in applying explainers to white-box models and black-box models.

-	The reason for using these three white box models is not very clear. Is there any way to verify their performance on real datasets?

-	It is better to analyze different explainers on real datasets, such as showing the actual meanings corresponding to some important features.

---

> ### Author Response · Authors · 2024-02-28
> **Response from the Authors**
>
> We would like to thank the Reviewer for their helpful comments, which have allowed us to improve our work. We next discuss the 4 weaknesses raised by the Reviewer.
>
> ###  **[W1] (Graph structure)**
>
> The proposed framework is aimed at assessing a given explainer on a very specific task, which is that of being able to detect “important” features that are hard-coded in our three white boxes. As we write in the **Limitations** part of Sec. 6, we acknowledge that this is just one specific test, and we do not pretend this to be the sole criterion for selecting among different explainers, nevertheless, it is important that explainers pass this test, to be considered reliable. As with any other GNN, our white boxes (except for the Local Model one, which is intentionally based only on the features of the node) blend features of neighboring nodes through the message-passing mechanism. Therefore, the graph structure plays a role in our NM and 2HM  white boxes as normal in GNN.
>
> ###  **[W2] (White boxes Vs. Black boxes)**
>
> The goal of our proposed framework is exactly to bypass the learning of the black boxes from synthetic training data with implanted logic, as typically done by other auditing frameworks in the literature. Such a training phase could introduce some approximation and uncertainty. In fact, there’s no guarantee that the injected logic is truly learned by the black-box model. We already mentioned (Sections 1 and 2) this as the main advantage of our more direct approach for assessing explainers for GNN-based classifiers, w.r.t. the methods in the literature. Finally, our truthful-to-the-model approach is totally orthogonal to this literature: one main contribution of our paper is showing that this different approach is possible and can produce interesting insights.
> We have improved the discussion about the difference with the methods in the literature in Sec. 1 and in Sec. 6. Please also refer to the response we gave to [W2] by Reviewer GwDM.
>
> ###  **[W3] (Performance on real datasets)**
>
> Please also see the response to [W2] of Reviewer GwDM. The process of auditing an explainer is inherently complex and cumbersome because the goal is to assess a software (the explainer) which in turn analyzes another software (the model), which in turn is the output of another software (the training algorithm). This process has many dependencies; for instance, it depends on the specific learning algorithm and its parameters. All these dependencies can make the overall auditing process unstable or even unreliable.  The reason for using the three white-box models is exactly to reduce the complexity inherent in the process, reducing the not-strictly needed dependencies. In our approach, by building the white boxes with known logic and hard-coded “important” features, we can get rid of the initial part of the process, eliminating the dependencies on the learning algorithm and its parameters, as well as the uncertainty about the logic which has been learned in the black-box model.
> Although our white boxes are not trained, our framework still needs data: i.e., the graph structure and the node features that constitute the GNN. In our experiments, we use both synthetic networks and a real-world dataset from Facebook social network. So yes, our framework can use real-world datasets.
>
> ###  **[W4]**
>
> As already said in response to [W3], we indeed use a real-world dataset. However, the semantics analysis of different explanation styles among explainers, or the actual meaning of important features, are not the focus of our benchmarking approach, in which instead we only test the explainers on their ability to detect the hard-coded “important” features as such.

---

### Review · Reviewer_GwDM · 2024-02-20

**Summary Of Contributions:**

This work considers the current literature on explainers for graph neural networks (GNNs): that is, algorithms that explain why a GNN made a specific prediction for a given input, usually by scoring the relevance of all input features, or in some cases identifying a subset of relevant nodes/edges/features. Many such approaches have been developed in recent years, and this work aims to provide a consistent means of comparison via a new benchmark. The idea of the benchmark is to design a couple simple GNN models, which have no learnable parameters and are completely specified in 1-2 layers, and test whether each explanation algorithm correctly identifies the set of "important" features. These important features are defined by the authors as those that can take on values that change the model's prediction when all other features are fixed. Under this evaluation, an explanation algorithm is deemed accurate if the important features have higher scores than the unimportant features.

The authors test five different graph explainers under their proposed approach. The results reveal certain insights about each method. The authors use these insights to provide some recommendations about how and when each method should be used.

**Audience:**

Yes

**Broader Impact Concerns:**

None.

**Claims And Evidence:**

No

**Requested Changes:**

Several questions and concerns are mentioned above. It would help to clarify these points in the paper.

**Strengths And Weaknesses:**

## Strengths

Explaining black-box models is a challenging problem with many different approaches, and GNNs seem to be particularly difficult. Certain metrics are widely used for tabular and image explanations, but it doesn't seem like there are reliable benchmarks for GNN explainers. It is therefore a worthwhile direction to explore, and the authors design a new approached anchored in a property that seems generally reasonable (although with some shortcomings described below). Their experiments consider a range of different methods, and are generally well executed and nicely presented.

## Weaknesses

Some concerns about the premise of this benchmark:

- Unlike explainers for tabular and image datasets, graph explainers have substantial variety in the format of their output. For example, some provide attribution scores for all input features, and others select important subsets of nodes or edges. Meanwhile, the benchmark tests methods via their ability to properly rank the important/unimportant features; many of the methods tested here are not designed to output single scores for each feature (i.e., scores that are not node-specific). It therefore seems that the benchmark provides a limiting and poorly matched way to evaluate and compare explanation approaches.
- The stated motivation for this "true to the model" benchmark (see the "Proposed framework" paragraph) is that previous evaluations are somehow not true to the model: they inject a dependency into the data, fit a black-box GNN, assume that the pattern is used by the model, and then test whether the explanation identifies it. It seems to me that the black-box model would in most cases use the injected pattern, do the authors believe that's not the case in the four cited works? Can they explain why or provide any evidence, because it seems like not the strongest motivation to design a new benchmark. Can the authors compare their findings to those works and explain whether they derive any new conclusions that are specifically enabled by their benchmarking approach, and not simply the inclusion of a different set of explainers?
- The definition of "important" features and axiom 1 seem a bit subjective, it's not obvious that these are desirable properties. For example, consider a GNN that classifies nodes based on the individual's `income` and `race`, so that the prediction is 0 if `income < X` or `race = Y`. If both conditions hold, neither feature would be deemed "important," and that seems like a bad thing. Are we sure this is a good definition of importance to base a benchmark on? Separately, it's worth contextualizing this view of "importance" via some existing notions in the literature, particularly the related axioms for SHAP and IntGrad (Lundberg & Lee, 2017 and Sundararajan et al, 2017).
- A smaller complaint about Axiom 1 is that the inequality between importance scores depends on the sign, but I'm not sure it should. For methods that provide signed importance scores (many feature attribution methods), an "important" feature should have a large negative score. So the axiom should if anything have $|\beta_f| < |\beta_{f^*}|$.
- Another complaint about Axiom 1: what is the relevance of this axiom for real models, where it seems very unlikely that any input feature is unable to have any effect on the model output?

About the white-box models and theoretical results:
- In the description of the white-box models, I couldn't follow the bit about considering the sequence $S$ of outputs of the base model for all nodes. I also couldn't follow why there was a softmax involved (mentioned in the Local Model section).
- Theorem 1 is a very intuitive result, I'm not sure it deserves to be called a "theorem." Maybe a claim, or a proposition instead. For all three models, the first direction is obvious. The second direction is also obvious from the contrapositive: if $\beta_f = 0$, the feature cannot change the output. These results shouldn't take 2 pages.

About the conclusions:
- Some of the insights from the experiments are known design choices of each method that have been thoroughly discussed in the literature. For example, the failure of some methods to identify important features when they're equal to the baseline value (e.g., zero); that some methods distinguish between positive and negative effects on the prediction; that some methods are sparse by design, whereas others identify all influential features. After reading the results, many of these claims seem like they would have been knowable without relying on the benchmark.
- Overall, the benchmark provides little information about each explanation's correction outside its assignment of "important" features. For example, it provides no way to evaluate the relative ranking of important features.

---

> ### Author Response · Authors · 2024-02-28
> **Response from the Authors [Part 1]**
>
> We would like to thank the Reviewer for their helpful comments, which have allowed us to improve our work.
>
> ### **[Strenghts]**
>
> We sincerely thank the Reviewer for recognizing the novelty of our approach and the quality of the execution.
>
> ### **[Weaknesses]**
> We next reply to all the weaknesses in the same order of the review, numbering them from 1 to 9.
>
> ### **[W1] (Scope of the proposal)**
> We agree with the Reviewer that our proposal is limited, or better said focused, on testing one specific aspect. Nevertheless, our test is rigorous and it is able to bring to light some important differences and pitfalls of different explainers. The simplicity of our proposal was already acknowledged and well discussed in the **Limitations** part of Sec. 6 of the first submitted version of the manuscript. We consider that part a good response to [W1]. Please refer also to our responses to [W2] and [W3].
>
> We also highlight that all the considered explainers, for what concerns features, output scores in the same format: i.e., as a function that takes the model, a node being classified, and returns a vector of "importance scores" for all features (see our [code](https://anonymous.4open.science/r/axiomatic-graph-xai/src/explainers.py)). We added a sentence clarifying this point in Sec. 4
> Anyway, even if the format of the feature explanations in node classification is the same in all explainers, we agree with the reviewer that there exists a considerable variety in the precise definition of "importance scores". These semantics are often gathered in the eyes of practitioners only by their intuitive meaning. The goal of the present work is to define a framework allowing to see through such differences.
>
> ###  **[W2] (Motivations)**
> Let us explain better the motivation at the basis of our endeavor. The process of auditing an explainer, as shown in the depiction below, is inherently complex and cumbersome, because the goal is to assess a software (the explainer) which in turn analyzes another software (the model), which in turn is the output of another software (the training algorithm). This process has many dependencies; for instance, it depends on the specific learning algorithm and its parameters. All these dependencies can make the overall auditing process unstable or even unreliable.  Moreover, there is a degree of uncertainty about whether the trained black box has learned the intended logic (i.e., the planted patterns).
>
> [Image](https://anonymous.4open.science/api/repo/axiomatic-graph-xai/file/fig1.png)
>
> As an exemplification of this, Longa et al. have to constrain their experiments only on black-boxes achieving > 95% accuracy with the injected patterns, which results in some explainers being dropped out of the assessment (see e.g., Table 2 and 3 in their paper). While this device is aimed at assuring that the classifier has learned the injected pattern, there is no formal guarantee that the logic learned by the black box truly matches the intended one. Instead, in our method the matching of the classifier logic with the intended one is formally guaranteed by Theorem 1.
>
> The idea at the basis of our proposal is exactly to reduce the complexity inherent in the process, reducing the not-strictly needed dependencies. In our approach, by building the white-boxes with known logic and hard-coded “important” features, we can get rid of the initial part of the process, eliminating the dependencies on the learning algorithm and its parameters, as well as the uncertainty about the logic which has been learned in the black-box model, obtaining a very stable test. As the task we want the explainer to accomplish is simple and clearly defined, if the test is not passed, we have a clear indication of a critical behavior by the explainer.
>
> Our truthful-to-the-model approach is thus complementary to the existing literature and can represent an interesting addition to the toolkit of existing methods to assess explainers for GNNs. The fact of doing something different from other recent proposals is not a drawback of our work, on the contrary, one key contribution of our paper is exactly showing that this different approach is possible and can produce interesting insights. We believe other researchers in the future might build on the same idea with more sophisticated white-boxes.
>
> We thank the Reviewer for pushing us to better highlight our motivations. We have added these considerations in Sec. 1 and in the Discussion in Sec. 6.

---

> ### Author Response · Authors · 2024-02-28
> **Response from the Authors [Part 2]**
>
> ###  **[W3] (Definition of “important” features)**
>
> The definition of “important” feature is by no means to be considered a general definition of a feature importance in machine learning. It is just the way we use, within the strict borders of our framework, to indicate the features that we would like an explainer to identify as important. These are the features that are hard-coded as “important” in the three white-box models with known internal logic. The goal of our benchmarking framework is to detect whether a given explainer is able to detect these features. The example of a classification based on a disjunction of two features, proposed by the reviewer, is perfectly legit and sound, simply it is beyond the goal of our framework. In fact, we do not learn any GNN from data where such a disjunction can emerge: our goal is just to assess a given explainer in its ability to detect those features that we can hard-code as “important” in our three white boxes. As we write in our Discussion in Section 6, we acknowledge that this is just one specific test, and we do not pretend this to be the sole criterion for selecting among different explainers, nevertheless, it is important that explainers pass this test, to be considered reliable. In other terms, we can say that passing our test is a necessary, but not sufficient, condition for an explainer to be considered a valid tool.
> We thank the Reviewer for highlighting that our presentation in the first version of the manuscript was not clear enough in these regards and could have confused the readers. For this reason, in the revised version we have addressed this concern by stressing more clearly what “important” really indicates and that by no means this must be considered as a general definition of a feature importance in machine learning. For the same reason, a comparison with the definitions of importance in the literature is not relevant. You can see the improved/added parts in blue in Sections 1, in Section 3 (before the definition of “important” features), and at the beginning of the Discussion in Section 6.
>
> ###  **[W4] (Axiom 1)**
>
> The Reviewer is right that for some explainers, the meaning of their importance scores follows an absolute value semantic where important features have a large magnitude in the scores that a given explainer produces, but with varying signs. However, this is not always the case (e.g. GraphLime). As such, we needed to establish an internally coherent notation: our framework assumes to have as input scores $\beta_f$ that are larger for importance features, and every aspect of our framework is coherent with this assumption. Explainers that follow different semantics need a bridge layer to make them coherent with our assumption. Therefore, some of the explainers indeed do have an absolute value function applied to their output before it is inputted to our model (line 35 of our [code](https://anonymous.4open.science/r/axiomatic-graph-xai/src/explainers.py)). We agree with the Reviewer that this detail was not properly highlighted in the previous version: we made it explicit in the new version (footnote 4 in Sec. 4).
>
> ###  **[W5] (Relevance to real models)**
>
> As discussed in the response to [W3], our definition of “important” features and the consequent Axiom, define a simple and clear test for the explainer under assessment. Can the explainer detect as important features that are hard-coded as important in the simple white-boxes with known logic? If an explainer fails this test, then this represents a criticality, regardless of the extent to which our white-boxes resemble real-world models. Our claim is that an explainer that fails our simple test is not a good choice for explaining real-world models or, at least, it requires further inspection of its usage and expected behavior.

---

> ### Author Response · Authors · 2024-02-28
> **Response from the Authors [Part 3]**
>
> ###  **[W6] (Description of the white boxes)**
>
> Thank you for your valuable feedback and for pointing out the confusion regarding the implementation details of our white-box models. We appreciate the opportunity to clarify these aspects of our work.
> Regarding your concern about the sequence $S$ of outputs from the base model for all nodes, we realize that our description might have been ambiguous. The sequence $S$ refers to the collection of output scores the model produces for each node in the graph. These scores represent the model's predictions based on the node features and their contextual information within the graph. We use the sequence $S$ of these outputs to dynamically adjust the midpoint $\vartheta$ and steepness $K$ of the sigmoid function. Specifically, we set $\vartheta$, the midpoint of the sigmoid, to the mean of $S$, and $K$, its steepness, to the inverse of the standard deviation of $S$. This adjustment serves as a normalization mechanism, ensuring that the rescaled output effectively has zero mean and unit variance. By doing so, we standardize the distribution of the base model’s outputs across different datasets and models, ensuring that on average, nodes have an equal probability of being classified into one class or the other, and that the variance in scores is consistent, facilitating comparability and interpretability across different modelling contexts.
> Concerning the mention of softmax in the Local Model section, we acknowledge this as an oversight in our manuscript. As correctly noted, our model applies a sigmoid function to the output scores to model binary outcomes in a node classification task. The mention of softmax was erroneous and not indicative of our actual implementation, which solely relies on the sigmoid function for the final output layer.
> We have revised Section 3.2 to clarify these points, and we hope the revised text addresses your concerns and clarifies the implementation and purpose of the parameters of the sigmoid.
>
> ###  **[W7] (Theorem 1)**
>
> Thank you for this comment. We agree that the proof of Theorem 1 is rather straightforward. However, this does not reduce the importance of the Theorem which is a cornerstone of our framework. For this reason, in the first submission, we decided to keep its (formal and detailed) proof in the main body of the paper. Now we realize it wasn’t a great presentation choice. We have thus moved its proof to the Appendix.
>
> ###  **[W8] (Insights of experiments)**
>
> There is a growing awareness concerning the necessity of explainability in a wide range of application domains, and consequently explainers are progressively included in a number of ML pipelines in industrial production. One cannot expect that these practitioners are interested in consulting the literature in order to understand the nuances of explainers they wish to use. While some of the reported insights might have been inferred by a thoughtful reading of the papers, we show how our benchmark could be used by practitioners to highlight undesirable effects of the machine learning and explainability pipeline.
> Furthermore, we argue that some of the pitfalls highlighted in our findings are not clearly stated in the original papers. The zero-encoding policy of LRP is mentioned in Bach 2015, but without a following discussion - to the point that is not included in the Captum implementation, the de-facto standard library for model interpretability and understanding in PyTorch. GNNExplainer follows a semantics (decision-aligned attribution) which is slightly different from all other explainers (signed attribution), and the impact of this distinction can only be grokked with a comparative literature review.. or our benchmark. We also highlighted how the constraint imposed by default by GNNExplainer to select a low number of important features often leads to a lower number of features recognized as important - another fundamental bit of information that can hardly be gained by reading the literature alone.
> Finally, the fact of “re-discovering” some known facts through new methods is usually considered a positive thing in research, and a proof of the validity of the proposed method.

---

> ### Author Response · Authors · 2024-02-28
> **Response from the Authors [Part 4]**
>
> ###  **[W9] (Relative ranking of important features)**
>
> Our approach is indeed based on a binary logic that overlooks the features’ ranking in the attributions: we already acknowledged and discussed this point in the **Limitations** part of Sec. 6. Our main argument is that being able to identify “important” features, even in our binary definition, is a necessary first step for assessing an explainer, and in this paper we show how this analysis is already sufficient to highlight pitfalls of, and differences amongst, state-of-the-art explainers. We do not claim that this approach constitutes an exhaustive criterion, and a number of more nuanced downstream analyses can be progressively carried out. Evaluating the relative rankings of important features is naturally a top topic for future investigation: generalizing the framework to handle the notion of non-binary importance of features is indeed the first time in the **Future work** part of Sec. 6.

---

> > ### Comment · Reviewer_GwDM · 2024-02-29
> > **Response**
> >
> > Thanks to the authors for responding to all the feedback. I'll quickly share a couple thoughts on the responses using the authors' numbering of issues.
> >
> > [W1] Line 34 of the [code](https://anonymous.4open.science/r/axiomatic-graph-xai/src/explainers.py) you shared illustrates the point I was making: these methods don't natively output scores in the format you're testing, which means you require an aggregation across nodes. Or am I missing something, it looks like that's what happens here? I recognize that it would more difficult to evaluate each method's per-node-per-feature scores, and it's not the worst thing to analyze each method this way, but it would help to clarify this point about the transformed output format.
> >
> > [W2] Thanks for clarifying, I think I understand the motivation. Pointing out Longa et al's decision to drop models with insufficient accuracy is interesting, and it looks like evidence of limitations to analyzing trained GNNs on data with synthetic dependencies (vs hardcoded GNNs). However, it also looks like they exclude only a small portion of the models, and the ones they're left with are all more realistic than the very simple models studied here. And once they've excluded the models with accuracy <95%, it seems like their claim that the model learned the synthetic dependencies is likely legitimate? And what about the other three papers (Rathee et al., 2022; Amara et al., 2022; Jaume et al., 2021), is there reason to doubt that their models learn the injected dependencies?
> >
> > Overall, I'll reiterate that based on the description in the introduction, the main distinction from previous works seems to be the use of models with hardcoded dependencies that guarantee reliance on specific features, vs learning realistic models on dataset with injected dependencies. Going this route limits the benchmark to very simple models, which is not ideal, so it would be nice to see some evidence that the models in previous papers really do fail to learn the injected dependencies. I don't think this is possible during the revision phase, so at a bare minimum it would be worth providing some reasoning for why they might fail to do so.
> >
> > And as requested in my review, it seems important to highlight differences in the conclusions from these various benchmarks. It would useful to know where the conclusions agree and what insights here are new. If I'm not mistaken, this discussion wasn't added during the revisions.
> >
> > [W3] Thanks for clarifying and adding the small update to Section 3.1. I think the definition of "important" you adopted is fine if it's explained properly, and it would help to caveat your definition immediately in Section 3.1, possibly by discussing an example like the disjunction. I would also disagree that comparing with existing notions of importance is irrelevant, it would help contextualize your proposal for readers that are familiar with but haven't memorized axioms from previous works.
> >
> > [W4] Thanks for clarifying the need to use the absolute value.
> >
> > [W5] It doesn't sound like you're arguing the axiom is relevant for real models? But I can agree it's a useful sanity check for the simple models studied here.
> >
> > [W6] Thanks for clarifying these points. Referring to the set of outputs as a "sequence" is a bit confusing.
> >
> > [W7] Moving the proof to the appendix seems like a good idea.
> >
> > [W8] Thanks for clarifying, I can agree that highlighting these nuanced implementation choices and their downstream effects is useful. It's probably also correct that the original papers don't always acknowledge potential unintended side-effects of such choices (e.g., the LRP zero-baseline) and practitioners would have to read many more papers to find thorough discussions. I'll maintain, however, that this a somewhat strange role for a benchmark to play. Benchmarks typically play a role of helping us appreciate the effects of difficult-to-reason about design choices via some metric that matters: e.g., using ImageNet to verify that residual connections make it possible to train deeper and better networks than previous works, or using the [insertion/deletion attribution metrics](https://arxiv.org/abs/1806.07421) to verify that RISE identifies influential image regions more effectively than GradCAM. In this case of this paper, the design choices exposed through the benchmark are knowable a priori, and that limits the impact.

---

> > > ### Author Response · Authors · 2024-03-01
> > > **Further responses from the Authors [W1]**
> > >
> > > We thank again the Reviewer for this engaging discussion which we deeply appreciate. Let us reply in the following to some of the latest comments by the Reviewer.
> > >
> > > ### **[W1]**
> > > We do indeed transform the explainers output. We explain this aggregation process in the framed box at the end of Section 3, specifically in these highlighted bullet points:
> > >
> > > > Run the explainer $\mathcal{E}$ on $\mathcal{M}$ for a set of test nodes $v \in V$, obtaining an importance score vector $\beta$, that should express (according to $\mathcal{E}$) the importance of each feature for the model $\mathcal{M}$ when classifying a node $v$.
> > >
> > > > For each test node, compute the ROC AUC between $\beta$ (the importance assigned by the explainer) and $\hat{\beta}$ (the ground truth importance of features for the white-box model $\mathcal{M}$). We call the mean of these measures the \emph{feature importance fidelity}.
> > >
> > > Besides explaining this process here and adding the reminder re-referencing this point in Section 4 as we did in the current revision, we do not see where else it would help to further clarify this point; we would be glad to accept any suggestion about where this point should be re-stated.

---

> > > > ### Author Response · Authors · 2024-03-01
> > > > **Further responses from the Authors [W2 - part 1]**
> > > >
> > > > ### **[W2]**
> > > > For the sake of motivating our work, we do not need to claim that either Longa et al or the other mentioned papers fail at learning the synthetic dependencies; simply, they do not provide **any formal guarantee** that this is the case, as instead, we do with our **Theorem**.  In the previous reply, we mentioned the practical device employed by Longa et al. in their experiments, just as an example of the practical issues involved in the process and to highlight that the whole process is very complex and heuristic and might depend on many factors and choices. The motivation for our work is exactly to reduce such complexity and dependencies, proposing a rather different approach from those in the literature. Isn’t this a good enough motivation?
> > > >
> > > > However, for the sake of positioning more clearly our contribution within this literature, let us discuss more in detail these related works and their limitations.
> > > >
> > > > The claim contained in the unpublished pre-preint by Longa et al (2022, arxiv) that the model has learned the synthetic dependencies cannot be taken for granted, even after discarding the <95%-accuracy classifiers. Correlations between features, spurious or naturally occurring, imply that there can be more than one way to achieve high accuracy. Therefore explainers should be evaluated on whether they reproduce the *specific* pattern picked by the classifier.
> > > > Let us assume that two features exhibit correlations in the data set (for instance, in the real data set we use, it could be the case that education level, birth year and location exhibit correlations). Depending on choice of the training set samples, shuffling, the level of noise in the labels, and other uncontrollable factors, a given classifier could be learning to give more importance to one feature or the other. Our point is that as a community we should make an effort into making sure that an explainer is able to reproduce exactly what has been learned by any given classifier, including simple ones. Instead, the impact of using a very complex classifier as an object to explain, while measuring the adherence of explainations to something different and much simpler (the original pattern), seems limited, and exploring alternatives seems essential.
> > > >
> > > > This point was also clearly stated in other papers cited by the reviewer (Rathee et al., 2022; Amara et al., 2022; Jaume et al., 2021): measuring explanations based on whether they are true to the data has clear limitations.
> > > >
> > > > Rathee et al. 2022 (Section 1) stated:
> > > >
> > > > _Some works employ synthetic datasets with an already-known subgraph (sometimes referred to as the ground truth reason or simply the ground truth). Explanations are
> > > > then evaluated based on their agreement with the ground truth. Such an evaluation is
> > > > sometimes flawed as one cannot always guarantee that the GNN has used in the first  place the seeded subgraph for its decision-making process (5). [...] The correctness of an explanation checks if the explainers is able to isolate spurious correlations and  biases that are intentionally added to the training data as a proxy for biases present in real-world data._
> > > >
> > > > In this paper the authors proposed different metrics, and only one of those is close to the goal of our paper, the so-called _correctness_. In this context, the authors artificially inject new edges to misclassified nodes, retrain a new classifier and measure how many of them are correctly identified by the explainer, thus pursuing a very different goal and result w.r.t. one proposed in our paper. Finally, interestingly enough, at the end of Section 3.3 about correctness Ratheer et al. wrote _“Note that our proposed approach of using injected correlations is different from using a synthetic graph with seeded ground truth. In particular, for seeded graph approach it is not always clear if the ground truth is actually picked up by the model to make its decision”_ - this is exactly the point we are making in proposing a white-box true-to-the-model approach.

---

> > > > > ### Author Response · Authors · 2024-03-01
> > > > > **Further responses from the Authors [W2 - part 2]**
> > > > >
> > > > > In Amara et al. 2022 the author reported in Section A.5 _“[...] the groundtruth explanations are artificially built and interpreted as the motifs which the nodes belong to. We are critical towards this method of assigning explanations as it is an a posteriori assignment and is only based on the labeling procedure. How we, humans, synthetically build and explain the node labels is not necessary the right explanation of the GNN model logic. The GNN might put its attention on different graph entities than the ones of the human-intelligible substructures. For this reason, we claim here that accuracy is not the right evaluation metric as it is limited to datasets where we have ground-truth explanations and in these very rare cases, we strongly question their "ground-truth" quality”_. Again, this is the same issue we are tackling in our work.
> > > > >
> > > > > Finally, Jaume et al. 2021 compare different explainability methods in a very specific setting (Computational Pathology). They consider three metrics to compare explainability methods. One of them considers as a “ground truth of explanations” the important features defined by expert pathologist and the ability to recover a good match of them by the explainer. The other two metrics are based on the idea that good explanations (i.e. identified explanatory features) have different distributions in the two classes targeted by the classification algorithm. While this is definitely a sound choice for the specific field of application (a clinical decision support system that has to be trusted and interpreted by clinicians), this claims can not be considered for more general benchmarking, where prior knowledge of explanations are not available or the separability between the two classes are more related to the idea of providing interpretable global explanation/model features importance, where important node features are able to separate the two classes rather than explaining single instance classifications.
> > > > >
> > > > > Incidentally, we also remark that, to the best of our knowledge,  Amara et al. 2022, Rathee et al. 2022, are – like Longa et al. 2023 – still pre-prints without peer-reviews: thus concurrently submitted to our work.
> > > > >
> > > > > We thank the Reviewer for making us survey these other pre-prints more carefully, and we are glad that also the authors of those papers clearly acknowledge the true-to-the-data approach limitations. If the Reviewer agrees with our discussion, we would be glad to add these considerations in the revised version. Due to the different nature of these benchmarks, a direct comparison with our findings is not obvious. In most of the cases they report performance of different explainers on different measures, without providing the type of insights we discuss in our findings. Nevertheless, in the revised version, in each explainer’s results subsesction (Sec. 5.1, 5.2, 5.3, 5.4, 5.5) we will add a short discussion of what was the performance of the explainer reported by other benchmarks and what insights were known facts.

---

> > > > > > ### Author Response · Authors · 2024-03-01
> > > > > > **Further responses from the Authors [W3]**
> > > > > >
> > > > > > We thank the Reviewer for the comment, and we agree that the introduction of our concept of ‘importance’ should be better conveyed if contextualized with similar metrics proposed in the literature. In particular, we will refer to the three papers by Rathee et al., 2022; Amara et al., 2022; Jaume et al., 2021., including the considerations stated in the argument [W2] above, respectively for the metrics of correctness, fidelity, and separability. Furthermore, we will include a discussion of the missingness and consistency properties as introduced in Lundberg et al., 2017. We will introduce these metrics in Section 2 of the paper and then add a comparison to our ‘importance’ concept after we have introduced it, in Section 3.1. It is worth stressing that the true-to-the-model approach that we are pursuing limits the comparability with true-to-the-data perspectives; we will expand this technical distinction in the paper, lest the user is tempted to equate these two approaches.

---

> > > > > > > ### Author Response · Authors · 2024-03-01
> > > > > > > **Further responses from the Authors [W8]**
> > > > > > >
> > > > > > > Although our benchmark can be seen as a sanity check (more than a metric) helping us appreciate the effects of "difficult-to-reason" design choices, it is not limited to exposing such “design choices”: one explainer could fail the test due to a variety of different reasons, not just because of “design choices”. Nevertheless, exposing design choices is a very useful feature when it comes to practitioners (not scientists) deploying XAI pipelines in production in an industrial setting.
> > > > > > >
> > > > > > > For what concerns the insights produced by our framework, it is not entirely the case that they are knowable a priori. For instance, for the case of GraphLIME, its failure to learn non-local features is in stark contrast to original claims about this explainer. As another example, in the case of GNNExplainer, even beyond the zero-value semantics, there is a discrepancy between the results obtained from this explainer and what was claimed, as highlighted by our sentence in Section 5.4:
> > > > > > >
> > > > > > > > _the presence of edges where varying a feature does not affect the output (even when there are edges where it does) makes it harder for this explainer to recognize a feature as important._

---

> > > > > > > > ### Comment · Reviewer_GwDM · 2024-03-01
> > > > > > > > **Response**
> > > > > > > >
> > > > > > > > [W1] I don't think you need to mention it in more places. The problem is the paper doesn't acknowledge that the output format you test (per-feature scores) *isn't native to any of the methods you consider*. If you think that's okay, feel free to explain why in the text, but the paper currently doesn't even acknowledge it. The following bullet point highlighted in your previous reply leaves out that the explainers don't by default provide per-node scores:
> > > > > > > >
> > > > > > > > > Run the explainer $\mathcal{E}$ on $\mathcal{M}$ for a set of test nodes $v \in V$, obtaining an importance score vector $\beta$, that should express (according to $\mathcal{E}$) the importance of each feature for the model $\mathcal{M}$ when classifying a node $v$.
> > > > > > > >
> > > > > > > > [W2] This is good, adding this discussion to the paper is a good idea.
> > > > > > > >
> > > > > > > > [W3] Sounds good, this will be an improvement.
> > > > > > > >
> > > > > > > > [W8] Sounds fine to me, I'm not going to recommend rejection based on this issue. Just something to think about that affects impact, you can discuss these points in the paper if you think it's helpful.

---

> > > > > > > > > ### Author Response · Authors · 2024-03-04
> > > > > > > > > **Yet another response from the Authors [W1]**
> > > > > > > > >
> > > > > > > > > Thanks again for this fruitful and engaging discussion.
> > > > > > > > >
> > > > > > > > > About [W1]:
> > > > > > > > > We are grateful to you for pointing out the lack of acknowledgment of the output format of the tested explainers. Indeed, many of the explainers do not natively provide per-node, per-feature scores. Reviewing our implementation, we can observe that only the explainers utilizing the Captum library are implemented to generate per-node, per-feature scores. Consequently, these scores must be transformed into a per-feature score format to facilitate comparison with other explainers (such as GraphLIME and GNNExplainer) that inherently produce aggregated per-feature scores. This harmonization to a per-feature score output is essential to ensure comparability across all explainers. Therefore, we must aggregate the per-node, per-feature scores from Captum-based explainers. In the revised version of our paper, we will better elaborate on this harmonization process and the necessary transformations.

---

### Author Response · Authors · 2024-02-28
**Revised version uploaded**

Dear AE and Reviewers,

Thank you for the time spent in managing and reviewing our work. We have already uploaded a revised version of the paper aimed at clarifying the motivations and contributions of our work, as well as other doubts raised by the reviewers.
To facilitate the review of the modifications, new content in the manuscript is highlighted in blue, while removed text is marked in red.
We next proceed to reply to the reviews, by means of a separate comment for each review.

Sincerely,
The Authors.

---

### Author Response · Authors · 2024-03-04
**Any further point for discussion?**

Dear AE, dear Reviewers,

We sincerely thank you for your effort in managing and reviewing our submission. We would like to ask the Reviewers, whether they are satisfied with our responses or if there is any further point to discuss in this phase.

Many thanks,
The Authors.

---

### Author Response · Authors · 2024-04-10
**Thanks!**

Dear AE, dear Reviewers,

We sincerely thank you for your effort in managing and reviewing our submission and for the many useful comments that helped us improving our work.

Many thanks,
The Authors.

---

### Decision · Action_Editor_K9AU · 2024-04-02

**Recommendation:** Accept as is

**Comment:**

During the discussion, two of the reviewers were leaning towards accepting the paper and one towards rejecting it. Overall, the reviewers tend to agree that the impact of the work may be limited since the proposed method is of limited in capacity and highly simplistic. However, as per TMLR guidelines, estimated potential impact should not be a criteria for the acceptance decision. Moreover, the reviewers also tend to agree that the problem tackled by this paper is highly relevant and should be studied more thoroughly in the literature. Hopefully, this paper will be first step towards a more rigorous evaluation of graph explainers. This is why I decide to recommend accepting the paper.

**Audience:**

Yes

**Claims And Evidence:**

Yes